# Diabetic Vasculopathy: Molecular Mechanisms and Clinical Insights

**DOI:** 10.3390/ijms25020804

**Published:** 2024-01-09

**Authors:** George Jia, Hetty Bai, Bethany Mather, Michael A. Hill, Guanghong Jia, James R. Sowers

**Affiliations:** 1Department of Medicine—Endocrinology and Metabolism, University of Missouri School of Medicine, Columbia, MO 65212, USA; george.jia@wustl.edu (G.J.); hbvvb@health.missouri.edu (H.B.); bjmb79@health.missouri.edu (B.M.); 2Department of Biology, Washington University in St Louis, St. Louis, MO 63130, USA; 3Department of Medical Pharmacology and Physiology, University of Missouri School of Medicine, Columbia, MO 65212, USA; hillmi@missouri.edu; 4Dalton Cardiovascular Research Center, University of Missouri, Columbia, MO 65212, USA

**Keywords:** insulin resistance, hyperinsulinemia, hyperglycemia, diabetes mellitus, diabetic vasculopathy

## Abstract

Clinical and basic studies have documented that both hyperglycemia and insulin-resistance/hyperinsulinemia not only constitute metabolic disorders contributing to cardiometabolic syndrome, but also predispose to diabetic vasculopathy, which refers to diabetes-mellitus-induced microvascular and macrovascular complications, including retinopathy, neuropathy, atherosclerosis, coronary artery disease, hypertension, and peripheral artery disease. The underlying molecular and cellular mechanisms include inappropriate activation of the renin angiotensin–aldosterone system, mitochondrial dysfunction, excessive oxidative stress, inflammation, dyslipidemia, and thrombosis. These abnormalities collectively promote metabolic disorders and further promote diabetic vasculopathy. Recent evidence has revealed that endothelial progenitor cell dysfunction, gut dysbiosis, and the abnormal release of extracellular vesicles and their carried microRNAs also contribute to the development and progression of diabetic vasculopathy. Therefore, clinical control and treatment of diabetes mellitus, as well as the development of novel therapeutic strategies are crucial in preventing cardiometabolic syndrome and related diabetic vasculopathy. The present review focuses on the relationship between insulin resistance and diabetes mellitus in diabetic vasculopathy and related cardiovascular disease, highlighting epidemiology and clinical characteristics, pathophysiology, and molecular mechanisms, as well as management strategies.

## 1. Introduction

Diabetes mellitus is a chronic disease characterized by hyperglycemia due to either insufficient production of insulin by the pancreas or the body not responding effectively to secreted insulin [1,2]. Data from the World Health Organization (WHO) show that the number of diabetic patients rose from approximately 108 million in 1980 to 422 million in 2014 [2]. More recent data suggest there are already some 463 million people with diabetes mellitus in the world, being the ninth leading cause of death with an estimated 1.5 million directly caused by diabetes mellitus in 2019 [2]. By 2030, it is expected that around 10.2% of the total world population or 578 million people will develop diabetes mellitus [3]. By 2045, this figure will increase to 10.9%, with the total number of diabetic patients reaching 700 million [3]. Specifically in the United States, more than 29 million Americans, representing approximately 10% of the US population, have diabetes mellitus [1,2]. The prevalence of diagnosed diabetes mellitus is highest among American Indians (14.7%), people of Latino origin (12.5%), and Blacks (11.7%), followed by Asians (9.2%) and Whites (7.5%) [1,2]. An increased prevalence of an unhealthy lifestyle, such as a Western diet characterized by over- and poor nutrition together with obesity, partially explains the global diabetes mellitus pandemic. Over time, the hyperglycemia and insulin-resistance/hyperinsulinemia associated with diabetes can lead to clinically evident microvascular and macrovascular complications in the eyes, kidneys, and nerves, as well as cardiovascular disease (CVD) characterized by coronary artery disease (CAD), heart failure, and cerebrovascular disorders. Therefore, a better understanding of the molecular mechanisms underlying diabetic vasculopathy in atherosclerosis, CAD, hypertension, and peripheral artery disease (PAD) may help in the development of novel approaches to prevent or delay the onset of diabetic vasculopathy. We briefly review the most current advances in the pathophysiology of diabetic vasculopathy and related CVD and highlight recent preventative and therapeutic strategies in diabetic patients.

## 2. Clinical Characteristics of Diabetic Vasculopathy in CVD

Type 1 diabetes mellitus (T1DM) is characterized by an absolute insulin deficiency induced by the T-lymphocyte-mediated autoimmune destruction of pancreatic β-cells [1,2]. As a result, the pancreatic β-cells cannot produce enough insulin to efficiently stimulate glucose uptake and maintain metabolic balance [1,2]. While T1DM typically tends to appear in childhood or adolescence, T2DM usually develops in those over the age of 40-years-old who are obese or overweight [1,2]. Both those with T1DM and T2DM present with diabetic vasculopathy. In T1DM as compared to T2DM, the correlation between hyperglycemia, microangiopathy, and macroangiopathy seems to be more significant [4,5]. Cardiovascular disease (CVD) mortality in patients with T1DM aged from 45 to 64 years at baseline increases by around 50% with each 1% increase in glycated hemoglobin (HbA1c) above normal values [4,5]. In a state of insulin resistance, the body’s tissues and cells respond insufficiently to insulin stimulation, leading to hyperglycemia. It has been shown that with increased visceral fat there is greater lipolytic activity leading to an increase in free fatty acids, which may repress insulin metabolic signaling and increase the risk for T2DM [1,2]. Patients with T2DM are at an increased risk of cardiovascular disease, partly owing to dyslipidemia. One clinical trial study from the Fenofibrate Intervention and Event Lowering in Diabetes (FIELD) has demonstrated that the lipid-lowering drug fenofibrate reduces total cardiovascular events, mainly due to fewer non-fatal myocardial infarctions and revascularizations without reducing the risk of the primary outcome of coronary events [6]. Epidemiological studies have demonstrated that insulin resistance is an independent risk factor for early diabetic vasculopathy. For instance, the Atherosclerosis Risk in Communities (ARIC) study found that individuals with glucose intolerance exhibit a greater risk for developing atherosclerosis and arterial stiffening than their counterparts with a normal glucose tolerance [7]. Meanwhile, interactions among elevated glucose, insulin, and triglycerides have a synergistic impact on arterial stiffening and play an important role in the early pathophysiology of diabetic vasculopathy in diabetic patients [7].

### 2.1. Atherosclerosis

The relationships between insulin resistance, T1DM, T2DM, and atherosclerosis are complex. Clinical data have found that patients with childhood-onset T1DM have a high prevalence of atherosclerosis characterized by necrotic plaque core formation, thrombosis, and severe arterial stenosis with increased macrophage and T-lymphocyte content [8]. Patients with T2DM are up to four times more likely than non-diabetic patients to develop atherosclerosis and CVD [9]. Epidemiological evidence supports an association between glycemic control and CVD risk. A meta-analysis based on four landmark clinical trials, the United Kingdom Prospective Diabetes Study (UKPDS), Action to Control Cardiovascular Risk in Diabetes (ACCORD), Action in Diabetes and Vascular Disease (ADVANCE), and Veterans Administration Diabetes Trial (VADT), suggested that intensive glucose control was associated with a 15% relative risk reduction in atherosclerosis and non-fatal/fatal myocardial infarction [10]. Recent data further indicated that an HbA1c between 6.0% and 6.9% was associated with low diabetes-mellitus-related mortality [9]. Cardiometabolic syndrome and related insulin resistance and hyperinsulinemia have been linked to atherosclerosis and CVD. Multiple studies have found that insulin resistance or hyperinsulinemia is a strong predictor of CVD [11,12]. The Insulin Resistance Atherosclerosis Study (IRAS) was the first clinical study to investigate the relationship between insulin resistance and CVD in a large multiethnic cohort [11]. This study verified that insulin resistance was an independent predictor of the increased risk of CVD in nondiabetic subjects during a follow-up period of 6.9 years after adjustment for confounding factors in glucose tolerance, fasting insulin, cholesterol, smoking, hypertension, and body mass index [11,12].

### 2.2. CAD

The impact of insulin resistance and diabetes mellitus on atherosclerosis is best documented in terms of its association with CAD. In a Finnish study, the presence of diabetes mellitus increased the seven-year risk of myocardial infarction and CVD-related death [13]. Moreover, the incidence of CVD was twice and three times in men and women with diabetes, respectively, than those nondiabetic patients [14]. In the Copenhagen City Heart Study, the relative risk of incident myocardial infarction was 2- to 3-fold higher in diabetic patients compared to nondiabetic individuals, and this increase was independent of hypertension and other CVD risk factors [15]. A recent meta-analysis from 4,549,481 individuals with T2DM showed that there was a 32.2% overall prevalence of macrovascular complications and CAD was the most frequently reported complication (21.2%) [16]. Moreover, diabetes mellitus also induces myocardial dysfunction (diabetic cardiomyopathy) in the absence of overt clinical CAD and other common CVD risk factors [1]. CAD events are more common and occur at a younger age in T1DM patients than in nondiabetic individuals [1]. Further, women with T1DM are more likely to have a CAD event than healthy women [1]. There is a difference in the pathology of CAD in patients with or without diabetes. For instance, arterial injuries in patients with T1DM present more severe stenoses and distal coronary findings involving multiple arteries than in those without diabetes [17]. An earlier autopsy study found that arterial plaques in T1DM are soft and fibrous and have a more concentric lesion location [18]. Another clinical study also demonstrated more obstructive and noncalcified lesions in patients with T2DM than in those with T1DM [19]. One multicenter, double-cohort, observational study has shown that the presence of diabetes is associated with an increased periprocedural risk of arterial stent restenosis without additional risk emerging during longer-term follow-up. This suggests that diabetes is a risk factor for early-stage in-stent restenosis in CAD [20].

### 2.3. Hypertension

There is a very high incidence of hypertension in patients with T1DM and T2DM. While patients with T1DM usually have microalbuminuria and clinical evidence of renal disease, most subjects with T2DM lack renal injury at the early stages of hypertension development. One study of 981 participants with T1DM showed that hypertension was present in 19% of patients with normoalbuminuria, 30% with moderately increased albuminuria, and 65% with macroalbuminuria [21]. The incidence of hypertension eventually reaches 75–85% in T1DM patients with progressive diabetic nephropathy [22]. Resistant hypertension is more common in patients with T1DM than in nondiabetic hypertensive individuals, and this resistance appears to be related to a higher risk for progression of diabetic nephropathy [23]. In contrast, most hypertensive patients with T2DM lack evidence of clinical renal disease. One clinical data set indicates that less than 20% of 3500 newly diagnosed T2DM patients have hypertension before the onset of moderately increased albuminuria [24]. In the Framingham Heart Study, T2DM was related to a 2- to 4-fold increased risk of hypertension, PAD, and myocardial infarction [25]. Another Framingham clinical study further showed that subjects who were hypertensive at the time of diabetes mellitus diagnosis had greater rates of all-cause mortality and CVD events compared with normotensive individuals with diabetes mellitus [26]. Insulin resistance has also been shown to increase the risk for hypertension. Approximately 50% of hypertensive patients are insulin resistant, and this defect in insulin action could contribute to the increased prevalence of both diabetic vasculopathy and hypertension [27]. An important CVD risk factor in diabetes mellitus and insulin-resistance-induced hypertension is arterial stiffening. For instance, patients with glucose intolerance or T2DM have greater arterial stiffness than their counterparts with normal glucose tolerance [7]. It is accepted that elevated glucose levels and inappropriate insulin secretion have a synergistic impact on arterial stiffening and play a key role in the early pathophysiology of hypertension and CVD in patients with T1DM and T2DM. Indeed, excessive arterial stiffness independently correlates with all-cause mortality and a composite end point of CVD in diabetic patients [28]. Additional important factors impacting the relationship between insulin resistance and diabetes-associated hypertension include race and gender. Thus, the Jackson Heart Study demonstrated that heightened insulin resistance is related to a greater risk of incident hypertension and progression towards elevated blood pressure levels among a Black population [29]. Interestingly, women with impaired glucose homeostasis and those with diabetes mellitus have a greater risk of hypertension than men with equivalent impairments in glucose tolerance and glucose homeostasis [30]. 

### 2.4. PAD

The increasing occurrence of insulin resistance and diabetes mellitus has implications for the prevalence and prognosis of PAD, an atherosclerotic occlusive disease of arteries in the lower extremities. Similar to CAD patients with diabetes, those with diabetes-related PAD also have a higher incidence of diffuse and complex atherosclerosis lesions and arterial disease than those of individuals without diabetes. PAD affects about 12 million Americans, and approximately 20–30% of these patients have diabetes mellitus [31]. A meta-analysis including 112,027 participants estimated an increase of 23.5% in the number of patients with PAD in the decade between 2000 and 2010 [32], which is the most common initial vascular manifestation in T2DM. In a recent cohort study of 1.9 million diabetic patients, 16.2% of diabetic patients presented with PAD as the first CVD manifestation [33]. Compared with men (14.4%), women had higher rates of PAD (26.6%) with an estimated prevalence of an ankle-brachial index ≤ 1.0, especially in low- and middle-income countries [34]. Concomitant diabetic peripheral neuropathy, which reduces sensory nerve feedback leading to a lack of pain perception, may predispose patients with diabetic PAD to present with ischemic ulcers and gangrene. PAD risk factors and related micro- and macrovascular comorbidity are very similar in T1DM and T2DM [35]. The basic pathophysiological changes in PAD initially present with early atherogenesis and gradually develop to the obstruction of and reduction in blood flow. While the pathophysiology of PAD in subjects with diabetes is similar to that of the non-diabetic individuals, diabetes is strongly associated with adverse outcomes following PAD revascularization [36,37]. Several pathogenetic mechanisms in the pathogenesis of PAD have been identified, including endothelial cells (EC) and vascular smooth cell (VSMC) dysfunction, inflammation, and increased platelet aggregation. 

## 3. Pathophysiology of Diabetic Vasculopathy

Both T1DM and T2DM increase the different risk of clinical events due to diabetic microvascular or macrovascular disease. Epidemiological studies have demonstrated an 11% to 16% increase in CVD events for every 1% increase in HbA_1c_ [38,39]. The Swedish National Diabetes Register data further provided evidence that elevated HbA_1c_ levels were a strong predictor of fatal and nonfatal CAD, stroke, and total mortality in a study of 18,334 patients with T2DM followed over a mean duration of 5.6 years [40]. Studies have further demonstrated that the risk of morbidity associated with diabetes is 2–3% for stroke, 2–5% for CAD, 20% for blindness, and 40% for PAD-related amputation [36,37]. Diabetic dyslipidemia has been implicated in atherosclerosis and diabetic vasculopathy. Related to this, in diabetes mellitus and insulin resistance elevated oxidized low-density lipoproteins (LDL) (oxLDL) are retained in and impair the function of the subendothelial layer of the vasculature inducing expression of ICAM-1 and VCAM-1, which further promote the infiltration and migration of circulating leukocytes through the endothelial wall into the media VSMCs. Excess uptake of oxLDL by macrophages also leads to the generation of foam cells, thereby contributing to the formation of atherosclerotic plaques (Figure 1). One of the important mechanisms responsible for hyperglycemia-induced diabetic vasculopathy is the nonenzymatic reaction between glucose and vessel proteins or lipoproteins. To this point, glucose initially forms early glycosylation products (Amadori products) with circulating or vessel proteins and subsequently rearranges to form advanced glycosylation end products (AGEs), which accumulate in the extracellular matrix (ECM) and contribute to diabetic vasculopathy [41]. AGEs bind and activate receptors for AGEs (RAGE) to increase the vascular production of reactive oxygen species (ROS) and induce the proinflammatory response. Molecule signal pathways, such as tumor necrosis factor alpha (TNF-α), nuclear factor kappa B (NF-κB), JNK, and interleukins (ILs), are involved in these pathophysiological processes (Figure 1) [41]. Indeed, studies have indicated that serum AGEs levels predict atherosclerotic plaque progression in diabetic patients with CAD [42,43]. Hyperglycemia is also accompanied by increased O-linked-N-acetylglucosaminylation (O-GlcNAcylation), which causes the posttranslational modification of vascular proteins. Persistent elevation of intracellular O-linked N-acetylglucosamine (O-GlcNAc) levels can induce chronic diabetic vasculopathy. For instance, excessive O-GlcNAc modification impairs vascular insulin metabolic signaling in the insulin receptor/insulin receptor substrate 1 (IRS-1)/phosphoinositide 3-kinases (PI3K)/protein kinase B (Akt)/endothelial nitric oxide (NO) synthase (eNOS) pathway, resulting in reduced NO production [44] (Figure 2). O-GlcNAc also inhibits the binding site of eNOS phosphorylation by Akt [44]. Moreover, O-GlcNAc alters the regulation of osteoprotegerin and osteocalcin and promotes arterial calcification in patients with diabetes mellitus [45]. 

Epidemiological evidence supports the notion that people with insulin resistance/hyperinsulinemia have impaired insulin metabolic signaling in vascular cells including ECs and VSMCs, thereby contributing to hypertension and diabetic vasculopathy (Figure 2) [46]. In this regard, our research data suggest that that obesity and over-nutrition enhanced the mammalian target of rapamycin (mTOR)/ribosomal S6 kinase 1 (S6K1) signaling pathway that induces vascular insulin resistance through the increased serine phosphorylation of the critical insulin-signaling/docking molecule insulin receptor substrate 1 (IRS-1) [46]. This serine phosphorylation further impairs IRS-1 tyrosine phosphorylation, PI3K engagement, and Akt phosphorylation/activation, as well as the downstream translocation of glucose transporter 4 (GLUT4) to the cell membrane and glucose uptake. Therefore, impaired insulin metabolic signaling via PI3K/Akt represses eNOS activation and NO production, which increase the phosphorylation of myosin light-chain kinase (MLCK), Ca^2+^ MLCK sensitization, leading to reduced VSMC relaxation, active smooth muscle contraction, and arterial stiffening (Figure 2). Meanwhile, hyperinsulinemia and vascular insulin resistance also activate the mitogen-activated protein kinase (MAPK)-dependent signaling pathway to increase the release of endothelin-1 (ET-1) which further promotes vascular constriction. Therefore, hyperglycemia and insulin-resistance/hyperinsulinemia induce atherosclerosis and hypertension, and thus increase the prevalence of diabetic vasculopathy.

Obesity is one of the predominant risk factors in developing insulin resistance and T2DM. Fat stored centrally, particularly within the abdomen (visceral fat), exhibits a much stronger association with insulin resistance, T2DM, and diabetic vasculopathy compared to fat stored in peripheral areas such as the gluteal or subcutaneous regions. Epicardial fat and perivascular adipose tissue have emerged as significant players in the development and progression of CAD [1]. Related to this, epicardial fat is a unique adipose deposit located between the myocardium and the visceral pericardium. Under physiological conditions, it serves various functions, including cushioning and protecting the heart [1,47]. However, the excessive accumulation of epicardial fat, especially in the context of obesity, has been associated with increased cardiovascular risk by locally releasing inflammatory adipokines [1,47]. Similar to epicardial fat, perivascular adipose tissue surrounds the coronary arteries and contributes to the development of atherosclerosis and CAD under the condition of obesity and diabetes [47].

## 4. Molecular Mechanisms of Diabetic Vasculopathy

### 4.1. Role of Renin Angiotensin–Aldosterone System (RAAS)

Inappropriate RAAS activation is an important contributor to diabetes mellitus and insulin-resistance/hyperinsulinemia-induced diabetic vasculopathy (Figure 3). Evidence from the Losartan Intervention For Endpoint (LIFE) reduction in hypertension [48], Heart Outcomes Prevention Evaluation [49], and the Antihypertensive and Lipid-Lowering Treatment to Prevent Heart Attack Trial (ALLHAT) [50] clinical trials indicates that RAAS antagonists decrease the incidence of new-onset T2DM. Data from the large prospective Impaired Glucose Tolerance Outcomes Research (NAVIGATOR) trial also demonstrated that valsartan, an angiotensin receptor blocker, caused a relative reduction of 14% in the incidence of diabetes mellitus in patients with glucose intolerance [51]. Our previous studies have also demonstrated that spironolactone, a mineralocorticoid receptor antagonist, prevented overnutrition-induced glucose intolerance, systemic and tissue insulin resistance, reduced NO, and arterial stiffening [52,53]. Recently, our group has further found that epithelial Na channels in ECs (EnNaC) are involved in these pathophysiological processes [54,55]. Related to this, epithelial Na channels are known to be a multimeric channel responsible for the maintenance of cellular Na^+^, and consequently cell water and volume homeostasis in several tissues including the kidney, lungs, and sweat glands. Our studies further found that ENaC is present in ECs and contributes to reduced bioavailable NO and vascular stiffening, which is related to reduced GLUT4 uptake and systemic and vascular insulin resistance [54,55]. Firstly, inappropriate RAAS activation induces serum and glucocorticoid-regulated kinase 1 (SGK1) activation that impairs EnNaC ubiquitination/degradation, resulting in EnNaC accumulation in the plasma membrane, and a net increase in Na^+^ channel activity. Elevated EnNaC expression and the membrane abundance in ECs increases Na^+^ influx and the polymerization of G-actin to F-actin, which further decreases EC eNOS activity, NO production, as well as arterial stiffening and microcirculation impairment [54,55]. Consistent with this notion, our recent research in obese mice indicates that EC-specific deletion of the alpha subunit of EnNaC prevents obesogenic-diet-induced reduced AMP-activated protein kinase α (AMPKα), sirtuin 1, and eNOS, as well as excessive vascular stiffness [54]. Inhibition of EnNaC with amiloride also decreases oxidative stress, endothelium permeability, inflammation, arterial fibrosis, and aortic stiffness, as well as cardiac diastolic dysfunction, without affecting blood pressure or Na^+^ retention [56,57].

### 4.2. Mitochondria Dysfunction and Excessive Oxidative Stress

Mitochondrial dysfunction, such as the loss of function in mitochondria and overproduction of oxidants, plays an important role in the development of impaired insulin metabolic signaling, insulin resistance, and associated diabetic vasculopathy (Figure 3). Diabetes-related mitochondrial dysfunction was initially found in the maternally inherited diabetes mellitus and deafness syndrome that presents with an A3243G mutation of mitochondrial DNA [58]. The loss of pancreatic beta cells also induces secondary mitochondrial dysfunction because of excessive cell apoptosis and the uptake of free fatty acids (FFAs) in T1DM and T2DM [1]. On the one hand, mitochondrial dysfunction decreases mitochondrial fatty acid oxidation, which in turn increases fatty acyl CoA and diacylglycerol levels and further inhibits insulin metabolic signaling and glucose uptake [1]. On the other hand, mitochondrial dysfunction also represses the hormone-sensitive lipase of adipocytes and endothelial lipoprotein lipase function, leading to the increased production of FFAs, abnormal lipid profiles, and related atherosclerosis and diabetic vasculopathy [1]. 

Mitochondrial dysfunction influences cellular bioenergetics in diabetic vasculopathy. The impaired capacity of mitochondria to efficiently generate adenosine triphosphate (ATP) compromises the energy balance within vascular cells, leading to vascular cellular energy deficits and associated vascular cell dysfunction [1]. One key mechanism through which mitochondria contribute to diabetic vasculopathy is excessive oxidative stress. Related to this, insulin resistance and diabetes mellitus are associated with the increased activation of NADPH oxidases, which are important sources of excess ROS production in the vasculature [1]. Almost all vascular cells, including ECs, VSMCs, and ECM, possess the ability to generate ROS. Excessive ROS production can lead to impairments in DNA, proteins, and lipids which in turn further contribute to mitochondrial dysfunction [1]. Meanwhile, increased ROS can also activate redox-sensitive signaling pathways to induce vascular inflammation, fibrosis, and remodeling [1]. Moreover, increased ROS are associated with reduced levels of bioavailable NO and EC dysfunction. Furthermore, disruptions in mitochondrial dynamics and autophagy contribute to the pathogenesis of diabetic vasculopathy [1]. Additionally, impaired mitophagy, a selective form of autophagy responsible for eliminating damaged mitochondria, allows the accumulation of defective mitochondria, further exacerbating mitochondrial dysfunction [1]. Therefore, mitochondrial dysfunction and oxidative stress are likely important instigators of diabetic vasculopathy.

### 4.3. Inflammation

Correlations between inflammation, diabetes mellitus, and CVD have been supported by extensive experimental and clinical evidence (Figure 3). For instance, C-reactive protein (CRP) levels are elevated in patients with insulin resistance, T1DM, and T2DM [59]. A proinflammatory state may occur in prediabetic individuals some 5 years before the actual onset of T2DM [60]. Patients with T2DM have elevated total leukocyte counts, particularly lymphocytes and neutrophils [61]. In T1DM, inflammation may be present in young patients, even soon after the diagnosis of T1DM. One study found that CRP is elevated within the first year of diagnosis of T1DM [62]. Circulating and tissue proinflammatory cytokines such as tumor TNF-α, interferon-γ, interleukin (IL)-6, and IL-12, vascular cell adhesion molecular 1, and monocyte chemoattractant protein-1 can impair insulin metabolic signaling and decrease insulin-mediated NO production, leading to arterial stiffening, hypertension, and other diabetic vasculopathy [1]. The promotion of T-helper-1 (Th1) responses is associated with M1-macrophage activation, which is generally related to an increase in pro-inflammatory responses [1]. In contrast, alternatively activated M2 macrophages, which express CD206, arginase-1, and IL-10, inhibit inflammation. Treg cells can improve insulin sensitivity by the inhibition of M1-macrophage polarization and increased M2-macrophage-mediated secretion of IL-10 [1]. Hyperglycemia and insulin resistance can activate the neuronal apoptosis inhibitor protein, leucine-rich repeat, and pyrin-domain-containing protein 3 (NLRP3) that mediates caspase-1 activation and the secretion of proinflammatory cytokines IL-1β and IL-18 and related diabetic vasculopathy [1]. 

### 4.4. Dyslipidemia

Dyslipidemia plays an important role in the development of CVD in patients with diabetes (Figure 3). The principle characteristics of dyslipidemia in T2DM include elevated fasting and postprandial triglycerides (TG), low high-density lipoproteins (HDL), elevated LDL, and a predominance of small dense LDL particles. In T1DM, elevated TG levels or hypertriglyceridemia may occur, but HDL levels are often normal or even elevated unless glycemic control is poor or diabetic nephropathy is present [46]. Insulin resistance is the primary mechanism leading to lipid metabolic disorders in subjects with diabetes [46]. Peripheral tissue insulin resistance increases the release of FFAs from adipose tissue, and elevated circulating FFAs levels are taken up by the liver where increased FFAs stimulate the synthesis of TG. Increased TG synthesis subsequently induces increased hepatic production of TG-rich very low-density lipoprotein cholesterol (VLDL) and the secretion of ApoB. TG-laden VLDL enrich LDL and HDL through the action of cholesterol ester transfer protein, leading to increased cholesterol in hepatocytes. These TG-rich LDL molecules are then hydrolyzed by hepatic or lipoprotein lipase that further increases the production of small dense LDL. Compared with those of non-diabetic individuals, patients with T2DM may not necessarily have a higher LDL level but usually have a greater abundance of small dense LDL particles, which are considered to be amongst the most powerful atherogenic components [16]. Reduced HDL concentrations are significantly related to patients with T2DM and therefore HDL is regarded as the “good” lipoprotein. Indeed, HDL can reverse cholesterol transport and have anti-inflammatory, anti-oxidative, and endothelium-dependent vasodilatory effects. Atherosclerosis involves the deposition of cholesterol-rich plaques on the arterial walls in diabetic vasculopathy. The chronic hyperglycemia or insulin resistance characteristic of diabetes enhances the inflammatory response within blood vessels, resulting in atherosclerosis [1]. Dyslipidemia also exacerbates this process by providing an abundant source of lipids for plaque formation [1]. Elevated LDL-C and triglycerides induce oxidative stress and inflammation in the vascular cells, promoting plaque instability and thus increasing the risk of cardiovascular events in diabetic vasculopathy [1].

### 4.5. Thrombosis

Diabetes mellitus and insulin resistance are regarded as risk factors for venous thromboembolism and pulmonary embolism (Figure 3). A meta-analysis from 63,552 patients with diabetes mellitus demonstrated a 1.4-fold increased risk for venous thromboembolism [63]. Further, it has been reported that the risk of venous thromboembolism in subjects with T1DM was 5.33-fold higher than in the non-T1DM group after adjusting for dyslipidemia, hypertension, and obesity [64]. Moreover, increased thromboxane B2 is related to platelet hyperactivity and coagulation in T2DM patients [65]. Under physiological conditions, insulin inhibits platelet aggregation and thrombosis by an increased fibrinolytic action and inhibition of tissue factor. However, hyperglycemia and insulin resistance facilitate platelet hyperactivity and atherothrombosis [66]. On the one hand, insulin resistance increases plasminogen activator inhibitor-1 (PAI-1) and fibrinogen while decreasing tissue plasminogen activator levels [66]. On the other hand, hyperinsulinemia induces tissue factor expression, procoagulant activity, and thrombin generation [66]. Therefore, plasma coagulation factors and lesion-based coagulants are increased while levels of endogenous anticoagulants are decreased in patients with insulin resistance and diabetes mellitus. These data suggest that a tendency for platelet activation and aggregation, accompanied by a propensity for coagulation, increases the risk of thrombosis, further complicating the progression of diabetic vasculopathy.

### 4.6. Endothelial Progenitor Cell (EPC) Dysfunction

Alterations in EPC function have been suggested to play key roles in diabetic vasculopathy (Figure 3). Related to this, EPCs, derived from bone marrow and peripheral blood, can differentiate into mature ECs and probably VSMCs, as well as produce growth factors, including vascular endothelial growth factor, fibroblast growth factor, and granulocyte-macrophage colony-stimulating factor that promote vasculogenesis and maintain vascular integrity [67]. Recent animal and clinical studies have demonstrated that the reduced numbers and weak function of EPCs are powerful markers of EC dysfunction in diabetes mellitus and related CVD. For instance, EPCs isolated from the peripheral blood of diabetic patients have exhibited EPC dysfunction pre-existing in those patients with CV risk factors prior to established CV disease [68]. The mechanisms underlying diabetes-mellitus-induced EPC dysfunction include attenuated bone marrow mobilization, reduced EPC proliferation, and shortened EPC survival, which decrease the circulating pool of primitive cells, inhibit vascular cell reparative capability, and promote diabetic vasculopathy [68]. 

### 4.7. Gut Dysbiosis

There is emerging evidence that diabetes mellitus and diabetic vasculopathy are associated with alterations in the gut microbiota, which include an estimated 100 trillion micro-organism species (Figure 3). Under normal conditions, the gut microbiota modulate the immune system, inflammatory response, and metabolic activities to contribute to overall homeostasis. However, risk factors such as obesity, insulin resistance, and diabetes mellitus induce dysbiosis of the gut bacteria, which has been implicated as a causative factor in CVD. Indeed, dysbiosis of the gut bacteria has been observed in insulin-resistant ob/ob [69] and db/db T2DM mice [70]. Clinical data further showed that microbial communities could exist in the arterial plaque and induce plaque instability, leading to CAD [71]. A causal contribution of gut microbiota dysbiosis to atherosclerosis susceptibility was first showed with the trimethylamine N-oxide (TMAO) as a gut-microbiota-derived factor, and the initial data demonstrated that the direct provision of TMAO accelerates atherosclerosis in murine models. Further supporting this, the inhibition of the gut microbiota-dependent conversion of nutrient precursors into TMAO represses choline-diet-induced atherosclerosis [72]. Moreover, metabolites derived from the intestinal microbiota are linked to the severity of myocardial infarction [73]. Recent data showed that there is a significant gut dysbiosis that results from decreases in microbial richness, diversity, and an increased Firmicutes/Bacteroidetes ratio in the spontaneously hypertensive rat and chronic angiotensin II infusion hypertensive rats [74]. Importantly, gut microbiota can produce norepinephrine which could directly promote vascular constriction and hypertension [75]. Moreover, Enterococcus faecalis can contribute to renal injury and hypertension by interfering with lipid metabolism [76]. Therefore, changes in fecal microbial community are associated with insulin resistance and diabetes-induced diabetic vasculopathy.

### 4.8. Extracellular Vesicles (EVs) and Their microRNAs (miRs)

EVs are small nano-sized vesicles that include microparticles, exosomes, and apoptotic bodies [1]. EVs mediate intercellular communication and regulate normal physiology and pathology [1]. A recent meta-analysis of 34 studies demonstrated that when compared to individuals without diabetes, patients with T2DM have higher levels of total circulating microparticles, which are derived from platelets, monocytes, and ECs [77]. Another cross-sectional study also suggested that diabetic patients have higher circulating levels of plasma EVs than euglycemic individuals [78]. Furthermore, increased circulating EC-derived EVs appear to be associated with atherosclerosis and vascular dysfunction in T2DM [79] (Figure 3). Interestingly, platelet-derived EVs also induce an increased release of EC-derived IL-1β, IL-6, and IL-8 and further promote vascular injury and atherosclerosis [80].

EVs contain molecular cargo, including mRNA, miRs, and long noncoding RNAs, DNA, proteins, and lipids [1]. The capacity to carry cell-specific information makes EVs attractive as biomarkers, because they may be indicators of diabetic vasculopathy and metabolic syndrome. Related to this, the large-scale miRNA profiling of plasma microparticles from 135 individuals with or without T2DM has demonstrated that miR-126 and miR-26a in circulating microparticles are significantly reduced in diabetic patients that are at higher risk for a concomitant CAD [81]. Further, the EV-derived miRNA-126/vascular endothelial growth factor 2 pathway was downregulated in untreated T2DM, potentially impairing vascular integrity [82]. Indeed, diabetic cardiovascular complications are associated with increased miR-223, -320, -501, and -504, as well as decreased miR-16, -133, -492, and -373 [83]. Whether these changes in miRNA are simply biomarkers of metabolic syndrome or whether they participate in EV-related diabetic vasculopathy remains to be clarified. 

## 5. Potential Preventive and Therapeutic Strategies for Diabetic Vasculopathy

In 2023, the American Diabetes Association (ADA) emphasized a comprehensive approach to managing diabetic vasculopathy, including the need to provide regular screening for vascular complications, maintaining tight control of blood glucose levels to reduce the risk of complications, controlling blood pressure, and managing lipid levels to prevent atherosclerosis and related complications, as well as recommending lifestyle modifications.

### 5.1. Lifestyle Management

Lifestyle modification, including smoking cessation; consumption of a healthy diet including vegetables, whole grains, fruits, and non-fat dairy products; regular physical activity; and weight reduction, can prevent/delay the progression of diabetes mellitus and CVD (Figure 3).

### 5.2. Glycemic Control

Insulin, metformin, and sulfonylureas are the first-line drugs for diabetes therapy (Figure 3). The UK Prospective Diabetes Study (UKPDS), a landmark randomized and multicenter trial in 5102 patients with T2DM, found that sulfonylurea or insulin therapy reduced the risk of the diabetes-related endpoint by 12%, and microvascular disease by 25% with a 16% trend towards a reduced risk of myocardial infarction [84]. In a recent meta-analysis including 40 studies and comprising 1,066,408 patients, it was shown that metformin reduced cardiovascular mortality, all-cause mortality, and CV events in CAD patients. Metformin has been shown to be more effective for reducing the incidence of CV events than sulfonylurea [85]. While sulfonylurea augments insulin secretion, metformin reduces blood glucose levels by inhibiting hepatic glucose production and promoting weight loss. Thiazolidinediones (TZDs) are peroxisome proliferator-activated receptor (PPARγ) agonists and have been widely used for treating patients with T2DM. Despite the increased risk of heart failure due to fluid retention, pioglitazone is consistently related to both a reduced risk of myocardial infarction and stroke, as well as atherosclerosis progression [86]. 

Recently, new antihyperglycemic agents, including the dipeptidyl peptidase 4 (DPP4) inhibitors, the glucagon like peptide-1 receptor (GLP-1R) agonists, and the sodium/glucose transporter 2 (SGLT2) inhibitors, have all been applied in diabetic patients with CVD. For instance, clinical trials in diabetic patients with a high cardiovascular risk of known CVD with a median follow-up of 1.5 to 3 years showed that DPP4 inhibitors reduced HbA1c by 0.2% to 0.36% without decreasing major adverse cardiovascular events [87,88,89,90,91]. Saxagliptin, unlike other DPP4 inhibitors, was associated with increasing the risk of heart failure [92,93]. GLP-1R agonists, including liraglutide, albiglutide, dulaglutide, and efpeglenatide, significantly reduced major adverse cardiovascular events by 12% to 27% [94,95]. Two meta-analyses have further shown that GLP-1R agonists reduced 12–13% of cardiovascular mortality, 12% of all-cause mortality, 6–9% of myocardial infarction, and 13–14% of stroke [96,97]. These GLP-1R agonists have demonstrated a reduction in major cardiovascular events, including cardiovascular death, nonfatal myocardial infarction, and nonfatal stroke in a diverse population of diabetic patients and these studies have contributed valuable insights into the cardiovascular safety and efficacy of novel antidiabetic medications [98]. Recent cardiovascular outcome trials have shown a cardiovascular benefit with SGLT2 inhibitors, which include dapagliflozin, canagliflozin, and empagliflozin. Related to this, a study of the Empagliflozin Cardiovascular Outcome Event Trial in Type 2 Diabetes Mellitus Patients (EMPA-REG) from 7020 patients with T2DM at a high risk for cardiovascular events found that empagliflozin reduced risk of the primary outcome, cardiovascular death, and all-cause mortality [99]. These benefits include a reduction in major adverse cardiovascular events, cardiovascular death, and hospitalization for heart failure. SGLT2 inhibitors have also shown a protective effect on renal outcomes and slowed the progression of kidney disease [100]. Empagliflozin not only inhibits renal glucose reabsorption, but also induces glycosuria and natriuresis, leading to a reduced blood volume and blood pressure. Empagliflozin also mitigates oxidative stress, inflammation, albuminuria, activation of the sympathetic nervous system, increases in uric acid levels, and endothelial dysfunction, fostering improved vascular function. Moreover, it demonstrates anti-fibrotic effects, suppressing cardiac remodeling. These multifaceted actions contribute to its remarkable ability to reduce major adverse cardiovascular events, particularly heart failure, making empagliflozin a pivotal therapeutic option beyond glycemic control in individuals with diabetes [99].

### 5.3. Cardiovascular Drugs

Timely and aggressive antithrombotic, lipid-lowering, antihypertensive treatments are warranted for both primary and secondary prevention in diabetic patients with CVD (Figure 3). The 2016 ADA Standards of Medical Care for Diabetes recommendations on aspirin therapy are consistent with the AHA and American College of Cardiology Foundation, and both suggest that low-dose aspirin should be considered in patients with an increased CV risk, as well as those at an intermediate risk [101,102]. The 2018 Cholesterol Guidelines recommend statins as the first-line therapy for both primary and secondary prevention in diabetic patients with CVD. The moderate-intensity statin should be used in T2DM with aging and the presence of CVD risk. Clinical guidelines in diabetic patients with hypertension have been derived from the widely accepted Seventh Report of the Joint National Committee on Prevention, Detection, Evaluation, and Treatment of High Blood Pressure (JNC 7), and the American Diabetes Association (ADA) have recommended strict treatments with RAAS and calcium blockades, aiming at values <130 mm Hg for systolic blood pressure and <80 mm Hg for diastolic blood pressure [103]. The recently revised ADA guidelines suggest that the blood pressure goal for individuals with diabetes and hypertension should be <140/80 mmHg [104]. Clinical randomized trials in the United Kingdom Prospective Diabetes Study (UKPDS), the Action to Control Cardiovascular Risk in Diabetes (ACCORD), Hypertension Optimal Treatment (HOT), Systolic Hypertension in Europe (Syst-Eur), and Systolic Hypertension in the Elderly (SHEP) demonstrated that strict blood pressure control has a beneficial role in diabetic patients with hypertension [105,106]. 

## 6. Conclusions

Diabetic vasculopathy is associated with CVD, including atherosclerosis, CAD, hypertension, and PAD. Hyperglycemia and insulin-resistance/hyperinsulinemia increase the prevalence of diabetic vasculopathy and related CVD. From a mechanistic point of view, inappropriate activation of the RAAS, mitochondrial dysfunction, excessive oxidative stress, inflammation, dyslipidemia, thrombosis, EPC dysfunction, gut dysbiosis, and the abnormal release of extracellular vesicles and their carrying miRs are involved in the development and progression of diabetic vasculopathy. Basic and clinical studies have suggested that the therapeutic strategies including lifestyle changes, glycemic control, antithrombotic therapy, and lipid-lowering and antihypertensive treatments have great benefits in the prevention of diabetic vasculopathy. The intensive application of these therapies has significantly reduced cardiovascular events and total mortality. However, the risk of adverse cardiovascular outcomes induced by intensive therapy remains significantly higher in individuals with diabetes than those without diabetes. Understanding the pathophysiology of insulin resistance and diabetes mellitus and associated vascular complications will facilitate the identification and development of novel treatments for the prevention and the treatment of diabetic vasculopathy.

## Figures and Tables

**Figure 1 ijms-25-00804-f001:**
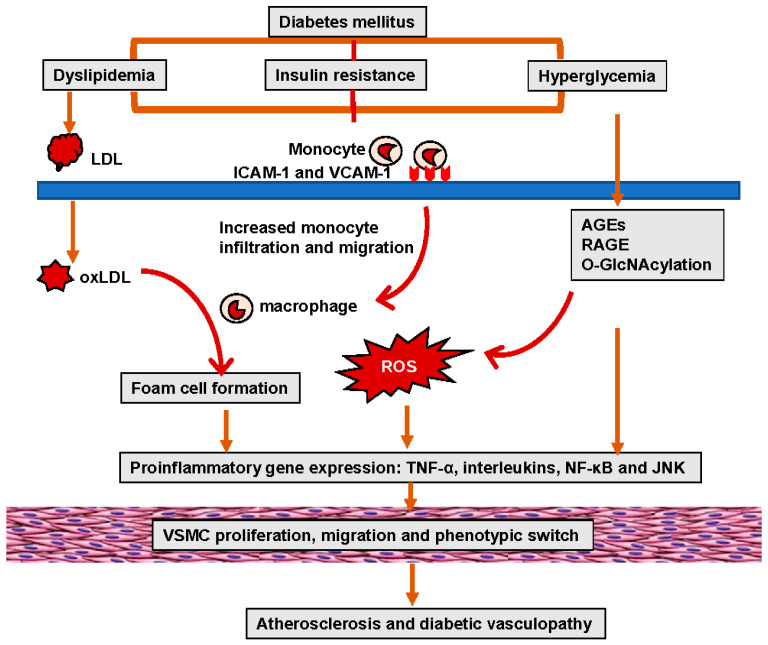
Hyperglycemia and insulin resistance in diabetes-mellitus-induced lipid disorders, oxidative stress, and inflammation, leading to EC and VSMC dysfunction, as well as atherosclerosis.

**Figure 2 ijms-25-00804-f002:**
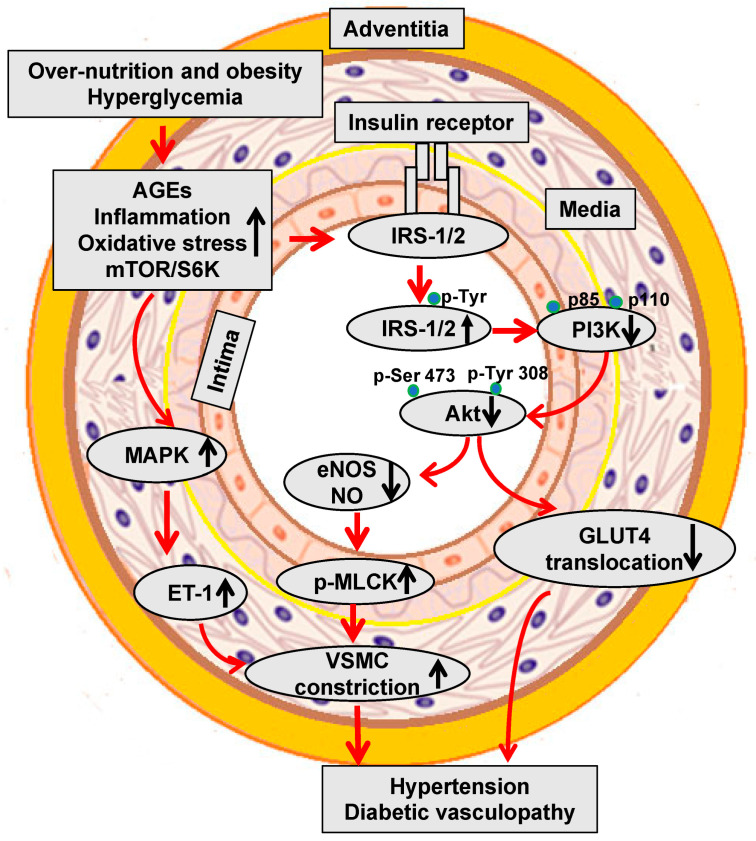
Hyperglycemia and insulin resistance impair vascular metabolic signaling and induce hypertension and diabetic vasculopathy.

**Figure 3 ijms-25-00804-f003:**
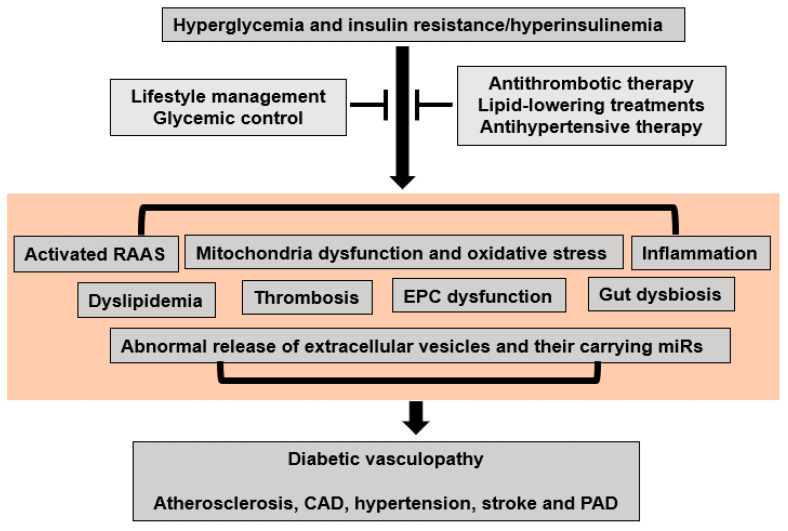
Proposed molecular mechanisms for hyperglycemia and insulin-resistance/hyperinsulinemia-induced diabetic vasculopathy.

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
