# Peer review of "Diabetic Vasculopathy: Molecular Mechanisms and Clinical Insights"

_ijms, 2024, doi:10.3390/ijms25020804_

Round 1
Reviewer 1 Report
Comments and Suggestions for Authors
This is review aimed to analyse the relationship between insulin resistance and diabetes mellitus in diabetic vasculopathy and related cardiovascular disease, highlighting epidemiology and clinical characteristics, pathophysiology and molecular mechanisms, as well as management strategies. The authors suggest that a better understanding of the molecular mechanisms underlying diabetic vasculopathy may help in the development of novel approaches to prevent or delay the onset of diabetic vasculopathy. The authors done the complex work analysing current and up-to-date data, in T1D as well as T2D. They discussed respected number of investigations and gave contribution for this topic in diabetes.
The issues to be resolved:
1. Please, add please point out why this review paper is different from the many others on this topic
2. Line 183. 3. Hyperglycemia and insulin resistance/hyperinsulinemia increase the prevalence of 183
diabetic vasculopathy, please use the appropriate new paragraph markup
3. Please discuss more references in the part 4.2 and 4.4
4. Please add more molecular mechanisms of cardiovascular protection of new innovative antihyperglycemic drugs, line 467-470
5. Please, add briefly data from DECLARE, LEADER, SUSTAIN6, PIONER 6, REWIND study
6. Please add briefly data from new studies with SGLT2i and GLP1RA showing cardiorenal protection (EMPAROR REDUCED and PRESERVED, DAPA HF, EMPA KIDNEY, DAPA CKD)
7. Please add briefly ADA recommendations 2023 in respect to presence/absence of diabetic vasculopathy
Author Response
Please see attached letter.

Reviewer 2 Report
Comments and Suggestions for Authors
A well-written paper.
It is an extensive narrative overview of the implications of T1DM and T2DM on the vascular system. It discusses molecular mechanisms, which fits nicely with the journal's topic. The language is very good and the paper reads well, in a structured manner.
I have only two clinical suggestions I would like to make:
- DM & CAD: emphasize the concept of DM being a factor of worse outcomes after coronary revascularization as well.
- DM & PAD:
1. Emphasize the concept of DM as a factor of prognosis different & individualized in different arterial districts (carotid, PAD, etc.) with different clinical events (vulnerable plaques in carotids, diffuse below-the-knee lesions in PAD).
2. Emphasize the concept of worse prognosis after PAD revascularization in diabetic patients (in-stent restenosis, etc.).
Author Response
Please see attached letter.

Reviewer 3 Report
Comments and Suggestions for Authors
The current review “Diabetic vasculopathy: molecular mechanisms and clinical insights” highlights the development and progression of diabetic vasculopathy. It is a very interesting review; I fully support the background to this work and recognize its need and topicality. However, I would like to make a few suggestions to further improve this review.
1) I suggest including this issue in the review “Clinical characteristics of diabetic vasculopathy in CVD.”
https://www.thelancet.com/journals/lancet/article/PIIS0140-6736(05)67667-2/fulltext.
Effects of long-term fenofibrate therapy on cardiovascular events in 9795 people with type 2 diabetes mellitus (the FIELD study): randomised controlled trial - The Lancet
2) Many clinical studies have shown significant associations between increased amounts of epicardial fat and coronary artery disease (CAD), it should be discussed.
Author Response
Please see attached letter.

Round 2
Reviewer 3 Report
Comments and Suggestions for Authors
The authors have done excellent job in reviewing the manuscript. I will recommend publication of this paper.